# Joint Design of Colocated MIMO Radar Constant Envelope Waveform and Receive Filter to Reduce SINR Loss

**DOI:** 10.3390/s21113887

**Published:** 2021-06-04

**Authors:** Liang Huang, Xiaofang Deng, Lin Zheng, Huiping Qin, Hongbing Qiu

**Affiliations:** 1School of Information and Communication, Guilin University of Electronic Technology, Guilin 541004, China; hliang@mails.guet.edu.cn (L.H.); cnzlin@guet.edu.cn (L.Z.); qiuhb@guet.edu.cn (H.Q.); 2State Key Laboratory of Integrated Services Networks, Xidian University, Xi’an 710071, China; 3Cognitive Radio and Information Processing Key Laboratory Authorized by China’s Ministry of Education Foundation, Guilin University of Electronic Technology, Guilin 541004, China; eephqin@scut.edu.cn; 4School of Electronic and Information Engineering, South China University of Technology, Guangzhou 510641, China

**Keywords:** colocated MIMO radar, constant envelope waveform design, receive filter, SINR, semidefinite programming, coordinate descent, similarity constraints

## Abstract

In this paper, we aim at the problem that MIMO radar’s target detection performance is greatly reduced in the complex multi-signal-dependent interferences environment. We propose a joint design method based on semidefinite relaxation (SDR), fractional programming and randomization technique (JD-SFR) and a joint design method based on coordinate descent (JD-CD) to solve the actual transmit waveform and receive filter bank directly to reduce the loss of strong interference to the output signal-to-interference-plus-noise ratio (SINR) of the radar system. Therefore, the maximization of output SINR is taken as the criterion of the optimization problem. The designed waveforms take into account the radar transmitter’s hardware requirements for constant envelope waveforms and impose similarity constraints on the waveforms. JD-SFR uses SDR, fractional programming and randomization technique to deal with the non-convex optimization problems encountered in the solution process. JD-CD transforms the optimization problem into a function of the phase of the waveform and then solves the transmit waveform based on CD. Compared with other methods, the proposed method has lower output SINR loss under strong power interference and forms deep nulls on the direction beampattern of multiple interference sources, which indicates that it has better anti-interference performance.

## 1. Introduction

Multiple-input multiple-output (MIMO) [1] radar has attracted the attention of many researchers in the past decades. MIMO radar has the ability to transmit different waveforms independently from each transmit antenna, whereas conventional phased array radars transmit only different phase-shifted versions of the same waveforms. Compared with the traditional phased array (PA) radar, MIMO radar has more degrees of freedom (DOF) [2] and can suppress more interference [3]. Especially, when the number of interferers increases, the output SINR of the phased array drops sharply [4].

MIMO radars can be divided into two types according to the arrangement of array spacing: distributed [5] and colocated [6]. The former has a large array spacing between the transmitter and receiver, and the signals transmitted from different antennas are relatively independent, which can be observed at multiple cross-sections of multiple targets, thus improving the performance of spatial diversity and azimuth estimation. The latter has small array element spacing, and all transmit antennas observe the same cross section of the scatterers. Each antenna transmits different waveforms to form a larger virtual aperture, which provides more flexibility for beampattern design [7] and higher interference suppression ability. In this paper, we use colocated MIMO radar as the research carrier of waveform design.

Waveform design is involved in improving the performance of radar in all aspects and has always been the focus of research [8,9,10,11,12,13,14,15]. The radar waveform design aims to focus the power on the target direction adaptively and to suppress the signal-dependent interferences from different directions in space to improve the output SINR. Waveform design can be divided into two broad categories. The first one considers only the waveform design of the transmitter [8,9,15,16,17,18]. For example, Imani and Ghorashi proposed two full-rank transmit covariance matrices ($R_{p_{1}}$, $R_{p_{2}}$) for the colocated MIMO radar [17], and the synthesized transmit waveform has the characteristics of constant envelope and low side lobe. One can increase the output SINR by suppressing interference from the direction. Additionally, the authors in [18] were inspired by [15] and [17] and proposed a robust waveform covariance matrix design method. The transmit covariance matrix has interference suppression, reduces SINR loss, and the waveform has a constant envelope characteristic. In [8,9,10,19,20], the waveform design of the transmit beampattern is studied. It should be noted that in a complex and multi-interference environment, this type of waveform design method that only optimizes the transmitter does not make full use of the degrees of freedom of the receiver [21,22]. The designed waveform and the output SINR are not optimal. Therefore, this approach is not considered in this article.

Second, the joint optimization of the radar transmitter and receiver is considered to solve the waveform design problem [14,23,24,25,26,27,28,29]. We will focus on this waveform design approach. The joint design focuses on suppressing the interference in different directions through the adaptive beamforming technique, giving full play to the advantages of the transmitter and the receive filter to process the interference, making the radar output SINR maximum. Since the objective function is usually a quadratic function, and the constraints imposed on the waveform are usually non-convex [1,2,3,5,7,11,16,17,24,30,31,32,33,34], it is difficult to solve the optimization problem [31]. Various methods [3,11,14,16,23,31,35,36,37] are used to solve the non-convex optimization problem. Feraidooni and Gharavian developed in [31] an algorithm for the joint design of a continuous/discrete [33,34,36] phase sequence and space-time receive filter to improve SINR, using the coordinate descent framework to deal with the constrained non-convex problem. Also, another joint design method is designed in [3]. Specifically, a design method of a joint transmit covariance matrix and receive filter is proposed to improve SINR in different situations, using Charnes–Cooper transforms to transform the established non-convex optimization problem into a new convex optimization problem. It is important to ensure that the designed waveform has the same transmit power through each antenna (constant envelope constraint). In addition, from the perspective of spectrum compatibility, the authors in [11] considered the constant envelope waveform design of MIMO radar spectrum compatibility in the presence of multiple targets and signal-dependent interference. Considering the constant envelope constraints and similarity constraints, and using SINR as the criterion, a non-convex optimization problem is established. In response to this problem, the authors proposed a phase-only dual ascent method (PDAM) to obtain the waveform.

In addition to using convex optimization to solve the waveform, there are other design ideas [22,38,39,40]. For instance, Zhou and Lu et al. introduce attenuation factors in [22] and define three sets of corner frequencies to formulate waveform design criteria to reduce the peak sidelobe level (PSL). In [38,39], the authors proposed a full-rank symmetric Toeplitz covariance matrix $R_{p_{m}}$ to design the transmit waveform, optimize the output SINR, and suppress unknown and unwanted sidelobe interference to reduce sidelobes. In another scenario, the authors in [40] proposed a joint design of the space-time transmit waveform and receive weight method, which detects all possible moving targets by setting a lower threshold. For all moving targets, the space-time transmit waveform and receive weights are updated iteratively through the proposed method.

In MIMO radar, due to parameter estimation [41,42,43,44], high-resolution range contour [45] and target detection [16,38,40,46,47,48] performance depends on the performance of the SINR. Therefore, it is very critical to study how to improve SINR in waveform design. Many researchers have performed this kind of work. For example, the authors in [42] proposes a direction of arrival (DOA) estimation of non-orthogonal MIMO radar based on reduced-dimension multiple signal classification (RD-MUSIC). In [47], considering the peak-to-average ratio (PAR) constraint, two non-decreasing algorithms are proposed to trade-off the output SINR and integrated sidelobe level (ISL). For multiple interactions between scatterers, the time-reversal multiple signal classification (TR-MUSIC) method were developed in [49], and in [48], the performance of this method was theoretically analyzed base on the mean-squared error (MSE) matrix of the position estimates. In addition, aiming at the coexistence of MIMO radar and a communication system, the authors in [43] propose an iterative algorithm based on the alternating direction method of multipliers (ADMM) [30] and block successive upper-bound minimization (BSUM) to obtain radar waveform and communication weight. In another study [32], under the influence of the digital radio frequency memory (DRFM) repetitive jammer, two waveform design criteria for suppressing the cross-correlation between the angle waveform and the interference are proposed to suppress the auto-correlation peak sidelobe level (APSL) and peak cross-correlation level (PCCL). Note that these two waveform design criteria are non-convex optimization problems. Aiming at this difficulty, the authors use the sequential quadratic programming (SQP) algorithm to solve.

However, in [14,19,20,23,24,29], the waveform optimization problem established adopts energy constraint, which cannot give full play to the amplifier’s maximum performance and reduce the output SINR. In [1,14,15,17,20,21,23,25,27,33,34,36], etc., pulse compression and ambiguity characteristics of the waveform are not taken into account, and the solution of [14,19,23] cannot guarantee global optimization. Therefore, the performance of the system can be improved through the design of the joint transmit waveform and receive filter based on semidefinite programming.

The rank of the waveform covariance in the colocated MIMO radar is the same as the output SINR as an important parameter because the rank represents the number of different transmitted waveforms. According to the research of [4], the MIMO radar waveform covariance matrix with more than two non-zero eigenvalues can output higher SINR than the phased array radar with rank 1. In multiple interference environments with strong power, if the phased array increases the number of antennas proportionally, the phased array can output a higher SINR than the MIMO radar. However, on a real battlefield, the number of radar antennas cannot be changed at will. Therefore, under the influence of multiple strong power interferences in different azimuths, the SINR output by the phased array radar drops sharply [3]. In [15], author Ahmed proposed a method that uses the cosine function with a step size of $\pi /N_{t}$ from 0 to π form a semi-positive definite Toeplitz matrix and covariance matrix generated by an auto-regressive process. Author Imani in [27] was inspired by [15], and author Imani proposed two algorithms based on joint design to design a full-rank waveform covariance matrix. It overcomes the shortcoming of [15] that the rank of the covariance matrix is only 2, and shows better anti-interference performance. However, since the covariance matrix generated by eigenvalue decomposition in first method is constrained by the fixed real set Ω, it cannot produce the optimal solution in [27].

Moreover, the designed waveform usually imposes some constraints on the waveform in actual engineering applications, for example, constant envelope constraint [3,8,9,10,11,12,16,18,21,27,30,32,34,35,36,38,50], PAR constraint [8,9,19,35,47], similarity constraint [11,16,29,43,50,51] or sidelobe constraint [15,21,22,30,32,33,38,47,50]. The constant envelope constraint is more stringent than the PAR constraint because most radar transmitters use nonlinear power amplifiers, which usually work in a saturated state, and the amplitude of the waveform is required to be constant. When the PAR exceeds the specified range, the waveform output from the amplifier is a reduced version, resulting in a significant reduction in the output gain of the matched filter. In addition to ensuring a constant envelope, the waveform also needs to have good waveform characteristics. It is well known that linear frequency modulation (LFM) signals are widely used in radar systems due to their good waveform characteristics [52,53]. Therefore, we can use similarity constraints to use LFM as a reference waveform to make the designed waveform share good waveform characteristics. The characteristics of the references are classified in Appendix A.

In this paper, we consider a detection environment with multiple strong signal-dependent interferences. Assuming the prior information of targets and interference is known [3,27,38], we design the transmit waveform and receive filter jointly to maximize the output SINR of the system as the optimization criterion, to reduce the large loss of the output SINR of the radar system during interference. Two joint waveform design algorithms are proposed to solve the actual constant envelope transmit waveform and receive filter bank. Moreover, constant envelope and similarity constraints are applied to the transmitting waveforms in this paper. In order to solve the non-convex quadratic optimization problem, JD-SFR introduces SDR, auxiliary matrix, and constant variables to correct the problem. Hence, the non-convex objective function can be transformed into an affine function. JD-CD is based on the coordinate descent method to decompose the waveform problem into a function of the phase of the waveform. Simulation results show that the proposed algorithm has a higher output SINR, and has more advantages in interference suppression than other methods.

The main advantages of the designed waveform and the contribution of this article are summarized as follows:Beampattern. Compared with the methods proposed in [23,47] and PA, the beampattern by our proposed methods has deeper nulls in the interference direction and good energy accumulation in the target azimuth. Moreover, when the interference number (or power) increases (no more than (DOF-1)), nulls can still be formed in the interference azimuth.SINR. The output SINR of our proposed methods is higher than the methods proposed in [23,47] and PA. Although the SINR loss of JD-CD is more than that of JD-SFR, the SINR output of JD-CD is still the highest, and the SINR loss is within an acceptable range.Waveform properties. The constant envelope waveform designed in this paper meets the actual hardware requirements of the radar transmitter and avoids the distortion of the waveform caused by the nonlinear effect of the amplifier. The good waveform characteristics of LFM are shared with the designed constant envelope waveform.The computational complexity of the proposed JD-CD is smaller than that of JD-SFR, and it reflects the process of waveform design concisely and intuitively, which can realize real-time waveform and receive filter design.

Notation: We adopt the notation of using boldface for matrices (upper case) and vectors (lower). ℂN×M are, respectively, the sets of N×M dimensional matrices of complex numbers. (⋅)∗, (⋅)T, (⋅)H and (⋅)−1 denote conjugate, transpose, complex conjugate transpose and inverse of matrix. IL denotes L×L dimensional identity matrix. E[⋅] denotes statistical expectation. |⋅| represents the modulus of a complex number. ‖⋅‖∞ denotes the Infinite norm of a matrix. j=−1 represents the imaginary unit. A≽0 means that A is a positive semidefinite matrix. tr(A), vec(A) and rank(A) represent the trace, vectorization and rank of matrix A. The Kronecker product and Hadamard product are represented by ⊗ and ⊙.

## 2. System Model

We consider a collocated narrowband MIMO radar system equipped with a transmit array of Nt and a receive array of Nr antennas as illustrated in Figure 1. The transmit and receive antennas are omnidirectional. Each transmit antenna sends a different waveform Sm(n),m=1,2,…,Nt,n=1,2,…L. L is the sample number in each waveform. The transmit waveform matrix of MIMO radar is expressed as S=[S(1),S(2),…,S(L)]∈ℂNt×L, S(n)=[S1(n),S2(n),…,SNt(n)]T∈ℂNt×1 represents the waveform emitted by Nt transmit antennas at time sample n. Then, the signal at target location θ is given by:(1)atT(θ)S(n),n=1,2,…,L.
where at(θ) represents the transmit steering vectors. For a uniform linear array (ULA) with d inter-element spacing, at(θ) has the following form:(2)at(θ)=1Nt[1,e−j2πdλsinθ,…,e−j2πdλ(Nt−1)sinθ]T∈ℂNt×1
where λ is the wavelength. Additionally, similar to at(θ), received steering vectors ar(θ) can be expressed as:(3)ar(θ)=1Nr[1,e−j2πdλsinθ,…,e−j2πdλ(Nr−1)sinθ]T∈ℂNr×1

The baseband signal at the receiver can be expressed as:(4)y(n)=x(n)+v(n)
where x(n) means K+1 echoes from different azimuths [3], v(n)∈ℂNr×1 describes the independent and identically distributed additive white Gaussian noise vector at Nr receivers with covariance matrix σv2I, namely, v(n)∼CN(0,σv2I).
(5)x(n)=∑i=0Kαmar(θm)atT(θm)s(n)
where αm is the radar cross section (RCS) of the mth scatterer. The received signal at the elements of the receive array will be stored in a Nr×L matrix Y as:(6)Y=∑m=0Kαmar(θm)atT(θm)S+V
where Y=[y(1),y(2),…,y(L)]Nr×L, S=[s(1),s(2),…,s(L)]Nt×L and V=[v(1),v(2),…,v(L)]Nr×L. Using equation vec(ABC)=(CT⊗A)vec(B), (6) can be rewritten as follows:(7)y=vec(Y)=∑m=0Kαm(IL⊗ar(θm)atT(θm))s+v
where s=vec(S) and v=vec(V). Assuming that there is a far-field target at azimuth θ0 and K signal-dependent interference sources are distributed at azimuth θi≠θ0,i=1,2,…,K [3], the received baseband signal is expressed as:(8)y=α0A(θ0)s+∑i=1KαiA(θi)s+v
where α0 and αi are the RCS of target and the ith interference sources, respectively. A(θ) is given by:(9)A(θ)=IL⊗ar(θ)atT(θ)

## 3. Problem Formulation

In this paper, in order to design a set of transmit waveforms that meet the constant envelope and similarity constraints, and to achieve the suppression of multi-strong signal-dependent interferences, we try to suppress multiple strong interferences in the radar airspace by placing null points in the interference azimuths of the transmit and receive beampattern, ensure the energy distribution of the target azimuth, and make the output SINR loss as little as possible in harsh environments. In other words, we maximize the output SINR to improve the target detection performance of the MIMO radar. 

### 3.1. Maximize Output SINR

In colocated MIMO radar, the SINR has a direct influence on the probability of target detection. Therefore, it is critical to increase the SINR. In the receiver, the signal passes through the matched filtering of the receive antenna and then outputs through the finite impulse response (FIR) filter w of length $N_{r}L\times 1$.
(10)$$\begin{array}{ll} \underset{s,w}{SINR} & =\frac{E[|\alpha_{0}w^{H}A(\theta_{0})s{|}^{2}]}{E[|\sum_{i=1}^{K}\alpha_{i}w^{H}A(\theta_{i})s{|}^{2}]+E[|w^{H}v{|}^{2}]}\\ & =\frac{E[|\alpha_{0}{|}^{2}\cdot |w^{H}A(\theta_{0})s{|}^{2}]}{\sum_{i=1}^{K}E[|\alpha_{i}{|}^{2}]\cdot |w^{H}A(\theta_{i})s{|}^{2}+E[|v{|}^{2}]\cdot |w^{H}{|}^{2}}\\ & =\frac{\rho_{0}|w^{H}A(\theta_{0})s{|}^{2}}{w^{H}R_{I}w} \end{array}$$
where $R_{I}$ is the covariance matrix formed by interference and noise $R_{I}=\sum_{i=1}^{K}\rho_{i}A(\theta_{i})ss^{H}A^{H}(\theta_{i})+I$, $\rho_{0}=\frac{E[|\alpha_{0}{|}^{2}]}{\sigma_{v}^{2}}$ and $\rho_{i}=\frac{E[|\alpha_{i}{|}^{2}]}{\sigma_{v}^{2}},i=1,2,\ldots ,K$.

### 3.2. Receive Filter Design

By jointly optimizing the transmit waveform and receive filter to maximize SINR, and assuming that the target and interference prior information is known, the signal of the interested azimuth can be output without distortion, while the interference and noise variance of the waveform output can be minimized. The famous minimum variance distortionless response (MVDR) method [54] can be utilized to optimize w for fixed s:(11)$$\left\{ \begin{array}{l} \underset{w}{\min}\;\;\;w^{H}R_{I}w\\ s.t.\;\;\;\;w^{H}A(\theta_{0})s=1 \end{array}\right.$$
Therefore, the optimal w for a fixed s is:(12)$$w=\frac{R_{I}{}^{-1}A(\theta_{0})s}{s^{H}A^{H}(\theta_{0})R_{I}{}^{-1}A(\theta_{0})s}$$

### 3.3. Constrained Waveform Design

In terms of analyzing the constraints of the waveform design, if the designed waveform only considers the application of energy constraints [14,19,20,23,24], the generated non-constant envelope waveform will cause the nonlinear distortion of the transmitter [23], and it is not ideal in some waveform characteristics. Therefore, this article requires that the modulus value of each element of the waveform must be a constant (i.e., constant envelope constraint), |s(k)|=c,k=1,2,…,NtL, where c=1/NtL represents the modulus of each transmit waveform normalized by the total emission energy. In addition, waveforms that only consider constant envelope or energy constraints often do not have good characteristics [28,51,55] (e.g., ambiguity characteristics and pulse compression). We use LFM signals [54] that are recognized to have good pulse compression and ambiguity characteristics as the reference waveform s0=vec(S0)∈ℂNtLx1. The LFM signal S0 expression is:(13)S0(m,n)=ej2πm(n−1)L⋅ejπ(n−1)2LNtL∈ℂNt×L,m=1,2,…,Nt;n=1,2,…,L
The designed transmit waveform s is similar to the reference waveform s0:(14)∥s−s0∥∞≤ξ, ξ∈[0,2]
where ξ represents the similarity parameter. When ξ=0 (i.e., s=s0), this means that the designed transmit waveform is completely similar to the reference waveform; when ξ=2, s and s0 are completely dissimilar and only satisfy the constant envelope constraint.

In this paper, our goal is to suppress multiple strong interferences and maximize the output SINR of the radar system, and the designed waveform meets the constraints. We can see from the expression of SINR that to maximize SINR, the interference and noise terms must be minimized, and noise is a random quantity of Gaussian white noise. Therefore, starting from the interference term, it can be seen that the size of the interference term depends on the receive filter w and the transmit waveform s. We use Equation (12) to solve w, and then further solve the following optimization problems to obtain the transmit waveform s. The optimization problem is expressed as:(15)$$\left\{ \begin{array}{l} \underset{s}{\max}\;\;\;\frac{\rho_{0}|w^{H}A(\theta_{0})s{|}^{2}}{w^{H}R_{I}w}\\ \;\;s.t.\;\;\;|s(k)|=\frac{1}{\sqrt{N_{t}L}},\\ \;\;\;\;\;\;\;\;k=1,2,\ldots ,N_{t}L,\\ \;\;\;\;\;\;\;\;∥s-s_{0}{∥}_{\infty }\leq \xi ,\\ \;\;\;\;\;\;\;\;\xi \in [0,2] \end{array}\right.$$
where s(k) is the kth element of s. As shown in problem (15), constraint $|s(k)|=1/\sqrt{N_{t}L},k=1,2,\ldots ,N_{t}L$ means that the amplitude of each element of the designed waveform is constant to meet the requirement of constant envelope. Constraint $∥s-s_{0}{∥}_{\infty }\leq \xi$ means that the maximum absolute value of the difference between the corresponding elements of the designed waveform and the reference waveform should be less than or equal to parameter ξ.

The designed waveform needs to meet the constant envelope constraint and is similar to the reference waveform $s_{0}$. The signal expression: (16)$$s(k)=\frac{e^{j\varphi_{k}}}{\sqrt{N_{t}L}},k=1,2,\ldots ,N_{t}L$$
Approximate by phase complete similarity constraints, that is:(17)$$\varphi_{k}=\arg s(k)\in [\gamma_{k},\gamma_{k}+\delta ]$$
where $\gamma_{k}=\arg s_{0}(k)-\arccos(1-\xi^{2}/2)$ and $\delta =2\arccos(1-\xi^{2}/2)$.

## 4. Proposed Algorithm

In this section, we propose two algorithms for the joint design of transmit waveforms and receive filter banks. JD-SFR is a joint constant envelope waveform and receive filter design method based on SDR, fractional programming and randomization technique; JD-CD is a joint constant envelope waveform and receive filter design method based on coordinate descent.

### 4.1. JD-SFR Algorithm

In order to transform the subsequent optimization problem (Algorithm 1), another equivalent form of the objective function SINR is:(18)$$\frac{\rho_{0}|w^{H}A(\theta_{0})s{|}^{2}}{w^{H}R_{I}w}=\frac{\rho_{0}s^{H}(A^{H}(\theta_{0})ww^{H}A(\theta_{0}))s}{s^{H}(\sum_{i=1}^{K}\rho_{i}A^{H}(\theta_{i})ww^{H}A(\theta_{i}))s+w^{H}ws^{H}s}$$
The optimization problem after equivalent transformation:(19)$$\left\{ \begin{array}{l} \underset{s}{\max}\;\frac{\rho_{0}s^{H}(A^{H}(\theta_{0})ww^{H}A(\theta_{0}))s}{s^{H}(\sum_{i=1}^{K}\rho_{i}A^{H}(\theta_{i})ww^{H}A(\theta_{i}))s+w^{H}ws^{H}s}\\ \;s.t.\;\;|s(k)|=\frac{1}{\sqrt{N_{t}L}},\\ \;\;\;\;\;\;\arg s(k)\in [\gamma_{k},\gamma_{k}+\delta ],\\ \;\;\;\;\;\;k=1,2,\ldots ,N_{t}L \end{array}\right.$$
problem (19) is non-convex, and it is very difficult to obtain the optimal solution. In order to obtain the optimal solution of the optimization problem (19), we introduce the matrix $R=s\cdot s^{H}$. The optimization problem in (19) can be rewritten as:(20)$$\left\{ \begin{array}{l} \underset{R}{\max}\;\frac{\rho_{0}tr(A^{H}(\theta_{0})ww^{H}A(\theta_{0})\cdot R)}{tr((\sum_{i=1}^{K}\rho_{i}A^{H}(\theta_{i})ww^{H}A(\theta_{i})+w^{H}w\cdot I)\cdot R)}\\ \;s.t.\;\;diag(R)=I,\\ \;\;\;\;\;\;R=s\cdot s^{H},\\ \;\;\;\;\;\;\arg s(k)\in [\gamma_{k},\gamma_{k}+\delta ],\\ \;\;\;\;\;\;k=1,2,\ldots ,N_{t}L,\\ \;\;\;\;\;\;rank(R)=1,\\ \;\;\;\;\;\;R≽0\; \end{array}\right.$$
Note that some constraints of problem (20) are non-convex. This type of optimization problem can be solved by semidefinite relaxation [55]. The rank 1 constraint and similarity constraints are temporarily discarded, and the optimization problem (20) is relaxed as:(21)$$\left\{ \begin{array}{l} \underset{R}{\max}\;\;\frac{\rho_{0}tr(A^{H}(\theta_{0})ww^{H}A(\theta_{0})\cdot R)}{tr((\sum_{i=1}^{K}\rho_{i}A^{H}(\theta_{i})ww^{H}A(\theta_{i})+w^{H}w\cdot I)\cdot R)}\\ \;s.t\;\;\;diag(R)=I,\\ \;\;\;\;\;\;\;R=s\cdot s^{H},\\ \;\;\;\;\;\;\;R≽0\; \end{array}\right.$$
where diag(R)=I and R≽0 are convex constraints. Unfortunately, the objective function is not a convex function about R. Therefore, the optimization problem is still not a convex optimization problem, and it is very difficult to find the optimal solution. Observation shows that the numerator and denominator of the objective function are linear. The objective function is a typical fractional programming problem [56]. We can use Charnes–Cooper to transform the optimization problem (21) into a positive semidefinite programming problem, let R=U/b:(22)$$\left\{ \begin{array}{l} \underset{U,b}{\min}\;\;tr((\sum_{i=1}^{K}\rho_{i}A^{H}(\theta_{i})ww^{H}A(\theta_{i})+w^{H}w\cdot I)\cdot U)\\ \;s.t\;\;tr(A^{H}(\theta_{0})ww^{H}A(\theta_{0})\cdot U)=1,\\ \;\;\;\;\;\;R=s\cdot s^{H},\\ \;\;\;\;\;\;diag(U)=b\cdot I,\\ \;\;\;\;\;\;U=R\cdot b,\\ \;\;\;\;\;\;U≽0,\\ \;\;\;\;\;\;b\geq 0 \end{array}\right.$$
where b means a positive scalar value, and U≽0 is a positive semidefinite matrix. For the optimization problem (22), the internal-point method [55] can be used to obtain the high-precision global optimal solution.

Obtain the global optimal solution Ropt=Uopt/bopt. The Gaussian randomization technique [57] can be utilized to obtain the optimal transmit waveform s of the original optimization problem (15). The main ideas are as follows: Generate an independent Gaussian random vector fim∈ℂNtL×1, subject to a mean value of 0 and a covariance of Q (i.e., fim∼N(0,Q),im=1,2,…,M). M is the number of randomization trials, where Q=Ropt⊙gcgcH∈ℂNtL×NtL, where gc=e−jγ/NtL∈ℂNtL×1, γ=[γ1,γ2,…,γNtL]T∈ℂNtL×1, and γk=args0(k)−arccos(1−ξ2/2). The imth randomization trial obtains the transmit waveform sim(k)=gc*(k)ψ(fim),im=1,2,…,M, where ψ(z)=exp{j(arg(z)⋅arccos(1−ξ2/2))/π}. Finally, the transmit waveforms of the M trials are, respectively, taken into the objective function (10) for calculation, and the transmit waveform that maximizes the SINR is taken as the optimal transmit waveform sopt. Bring sopt into Equation (12) to update wopt, and then update sopt with the latest wopt until the adjacent SINR change value is less than the expected value, stop iterating and output the final transmit waveform sopt and receive filter wopt.
**Algorithm 1.** JD-SFR**Input**: Nt,Nr,M,θ0,α02,{αi2,θi}i=1K,αv2,ξ.**Output**: sopt and wopt0: Set n=0, initialize the transmit waveform s(n)=s0 (use Equation (13) to calculate the reference waveform S0, s0=vec(S0)) and the receive filter w(n) is calculated by (12). Then, compute SINR(n) by Equation (10).1: n=n+12: Compute U(n) and b(n) using the problem in Equation (22), respectively3: R(n)=U(n)/b(n)4: Compute s(n) by Gaussian randomization technique3: RI(n)=∑i=1KρiA(θi)s(n)s(n)HAH(θi)+I4: Compute w(n) by Equation (12)5: Compute SINR(n) by Equation (10)6: If |SINR(n)−SINR(n−1)|≤η, where η is a user selected parameter to control convergence, stop iterating the output optimal transmit waveform sopt=s(n) and receive filter wopt=w(n). Otherwise, repeat step 1 until convergence.

In this section, we propose JD-SFR to jointly design the transmit waveform and receive filter to reduce the influence of interference on the target detection performance of the MIMO radar system and improve the output SINR. We also consider two important practical constraints for the designed waveforms: the constant envelope constraint and the similarity constraint. The established non-convex optimization problem is solved by the SDR method and fractional programming transformation, and the high-precision approximate solution of the original waveform optimization problem is obtained by the randomization technique. 

We now analyze the computational complexity of the JD-SFR algorithm. The overall complexity of JD-SFR is linear with the number of iterations. In order to facilitate understanding, we specifically analyze the main computational complexity of the nth iteration. The complexity of JD-SFR mainly comes from solving problem (22) ($O({\left(LN_{t}\right)}^{3.5})$) and randomization ($O(M{\left(LN_{t}\right)}^{2})$). The reason is that the complexity of each randomization calculation is $O({\left(LN_{t}\right)}^{2})$. Therefore, the computational complexity of JD-SFR is $O(n_{o}{\left(LN_{t}\right)}^{3.5})+O(n_{o}M{\left(LN_{t}\right)}^{2})$, where $n_{o}$ represents the number of iterations required to obtain $s^{opt}$ and $w^{opt}$. Note that the disadvantage of JD-SFR is that the calculation process is more complicated. In the next section, we will introduce a concise algorithm.

### 4.2. JD-CD Algorithm

Aiming at the relatively complex calculation process of JD-SFR, we propose another algorithm for the joint design of a constant envelope waveform and receive filter based on coordinate descent. JD-CD optimizes the phase of each element of the transmitted waveform one by one and updates it in time. Each calculation is based on the waveform optimized in the previous step for further optimization iterations. (Algorithm 2)

In order to facilitate the individual calculation of the following waveform elements, we will convert Equation (18) into an expression about the waveform s:(23)$$\begin{array}{ll} \frac{\rho_{0}|w^{H}A(\theta_{0})s{|}^{2}}{w^{H}R_{I}w} & =\frac{\rho_{0}s^{H}(A^{H}(\theta_{0})ww^{H}A(\theta_{0}))s}{s^{H}(\sum_{i=1}^{K}\rho_{i}A^{H}(\theta_{i})ww^{H}A(\theta_{i}))s+w^{H}ws^{H}s}\\ & =\frac{\rho_{0}s^{H}(A^{H}(\theta_{0})ww^{H}A(\theta_{0}))s}{s^{H}((\sum_{i=1}^{K}\rho_{i}A^{H}(\theta_{i})ww^{H}A(\theta_{i}))+w^{H}w\cdot I)s}\\ & =\frac{s^{H}R_{num}s}{s^{H}R_{den}s} \end{array}$$
where
(24)$$R_{num}=\rho_{0}A^{H}(\theta_{0})ww^{H}A(\theta_{0})$$
And
(25)Rden=(∑i=1KρiAH(θi)wwHA(θi))+wHw⋅I
The optimization problem (15) can be rewritten into the following form:(26){maxs   sHRnumssHRdens  s.t.   |s(k)|=1NtL,        k=1,2,…,NtL,        ∥s−s0∥∞≤ξ,         ξ∈[0,2]
where s=[s(1),s(2),…,s(k),…,s(NtL)]T∈ℂNtL×1 represents the transmit waveform vector, and s(k) represents the kth interval element of the transmit waveform.

In order to optimize each waveform element, the objective function (23) is decomposed into a function with respect to the phase of s(k). This is conducive to taking advantage of the coordinate descent algorithm that optimizes only one element at a time. The numerator $s^{H}R_{num}s$ of the objective function can be expressed as:(27)$$\begin{array}{ll} s^{H}R_{num}s & =R_{num}(k,k)\cdot |s(k){|}^{2}+\sum_{\begin{array}{l} h=1\\ h\neq k \end{array}}^{N_{t}L}\sum_{\begin{array}{l} z=1\\ z\neq k \end{array}}^{N_{t}L}s^{*}(h)R_{num}(h,z)s(z)+\left(2\sum_{\begin{array}{l} h=1\\ h\neq k \end{array}}^{N_{t}L}s^{*}(h)R_{num}(h,k)\right)\cdot s(k)\\ & =R_{num}(k,k)\cdot c^{2}+\sum_{\begin{array}{l} h=1\\ h\neq k \end{array}}^{N_{t}L}\sum_{\begin{array}{l} z=1\\ z\neq k \end{array}}^{N_{t}L}s^{*}(h)R_{num}(h,z)s(z)+\left(2\sum_{\begin{array}{l} h=1\\ h\neq k \end{array}}^{N_{t}L}s^{*}(h)R_{num}(h,k)\right)\cdot c\cdot e^{j\cdot \varphi_{k}}\\ & =r_{n1k}+r_{n2k}\cdot e^{j\cdot \varphi_{k}} \end{array}$$
where $R_{num}(i_{row},i_{col}),i_{row}=1,2,\ldots ,N_{t}L,i_{col}=1,2,\ldots ,N_{t}L$ represents the $\left(i_{row},i_{col}\right)$th element of matrix $R_{num}$, and c is the amplitude of each element of the waveform. In the above numerator expression, in order to make the subsequent calculation process intuitive and make the waveform meet the constant envelope, we directly apply the constant envelope constraint |s(k)|=c to the waveform elements:
where
(28)$$r_{n1k}=R_{num}(k,k)\cdot c^{2}+\sum_{\begin{array}{l} h=1\\ h\neq k \end{array}}^{N_{t}L}\sum_{\begin{array}{l} z=1\\ z\neq k \end{array}}^{N_{t}L}s^{*}(h)R_{num}(h,z)s(z)$$
And
(29)$$r_{n2k}=2c\sum_{\begin{array}{l} h=1\\ h\neq k \end{array}}^{N_{t}L}s^{*}(h)R_{num}(h,k)$$
The denominator $s^{H}R_{den}s$ of the objective function is the same as the numerator, which can be written as:(30)$$\begin{array}{ll} s^{H}R_{den}s & =R_{den}(k,k)\cdot |s(k){|}^{2}+\sum_{\begin{array}{l} h=1\\ h\neq k \end{array}}^{N_{t}L}\sum_{\begin{array}{l} z=1\\ z\neq k \end{array}}^{N_{t}L}s^{*}(h)R_{den}(h,z)s(z)+\left(2\sum_{\begin{array}{l} h=1\\ h\neq k \end{array}}^{N_{t}L}s^{*}(h)R_{den}(h,k)\right)\cdot s(k)\\ & =R_{den}(k,k)\cdot c^{2}+\sum_{\begin{array}{l} h=1\\ h\neq k \end{array}}^{N_{t}L}\sum_{\begin{array}{l} z=1\\ z\neq k \end{array}}^{N_{t}L}s^{*}(h)R_{den}(h,z)s(z)+\left(2\sum_{\begin{array}{l} h=1\\ h\neq k \end{array}}^{N_{t}L}s^{*}(h)R_{den}(h,k)\right)\cdot c\cdot e^{j\cdot \varphi_{k}}\\ & =r_{d1k}+r_{d2k}\cdot e^{j\cdot \varphi_{k}} \end{array}$$
where $R_{den}(i_{row},i_{col}),i_{row}=1,2,\ldots ,N_{t}L,i_{col}=1,2,\ldots ,N_{t}L$ represents the $\left(i_{row},i_{col}\right)$th element of matrix $R_{den}$,

Where
(31)$$r_{d1k}=R_{den}(k,k)\cdot c^{2}+\sum_{\begin{array}{l} h=1\\ h\neq k \end{array}}^{N_{t}L}\sum_{\begin{array}{l} z=1\\ z\neq k \end{array}}^{N_{t}L}s^{*}(h)R_{den}(h,z)s(z)$$
and
(32)$$r_{d2k}=2c\sum_{\begin{array}{l} h=1\\ h\neq k \end{array}}^{N_{t}L}s^{*}(h)R_{den}(h,k)$$
Rewrite problem (26) in the form of a phase:(33)$$\left\{ \begin{array}{l} \underset{\varphi_{k}}{\max}\;\;\;\frac{r_{n1k}+r_{n2k}\cdot e^{j\cdot \varphi_{k}}}{r_{d1k}+r_{d2k}\cdot e^{j\cdot \varphi_{k}}}\\ \;\;s.t.\;\;\;\varphi_{k}\in [\gamma_{k},\gamma_{k}+\delta ],\\ \;\;\;\;\;\;\;\;k=1,2,\ldots ,N_{t}L \end{array}\right.$$

Problem (33) is the optimized design of a continuous transmit waveform. JD-CD is also suitable for the optimized design of finite alphabet transmit waveforms in discrete cases. The assumed waveform element is $s(k)\in c\{e^{j\frac{2\pi }{M_{d}}\cdot 0},e^{j\frac{2\pi }{M_{d}}\cdot 1},\ldots ,e^{j\frac{2\pi }{M_{d}}\cdot (M_{d}-1)}\}$. Among them, $M_{d}$ represents the number of discrete alphabets. At the same time, the reference waveform is $s_{0}(k)\in c\{e^{j\frac{2\pi }{M_{d}}\cdot 0},e^{j\frac{2\pi }{M_{d}}\cdot 1},\ldots ,e^{j\frac{2\pi }{M_{d}}\cdot (M_{d}-1)}\}$. According to the definition of infinite norm, $∥s-s_{0}{∥}_{\infty }=\max_{k\in \{1,2,\ldots ,N_{t}L\}}|s(k)-s_{0}(k)|\leq \xi$. After equivalent transformation $ℜ[s(k)s^{*}(k)]\geq 1-\xi^{2}/2$, where ℜ[ω] represents the real part of ω.

The phase expressed by Equation (17) is further expressed in detail as:(34)$$\varphi_{k}=\arg s(k)\in [\arg s_{0}(k)-\arccos(1-\xi^{2}/2),\arg s_{0}(k)+\arccos(1-\xi^{2}/2)]$$
Discrete the value interval of s(k) into an expression form containing a finite alphabet:(35)$$s(k)\in c\{e^{j\frac{2\pi }{M_{d}}\cdot p_{k1}},e^{j\frac{2\pi }{M_{d}}\cdot p_{k2}},\ldots ,e^{j\frac{2\pi }{M_{d}}\cdot p_{kx}},\ldots ,e^{j\frac{2\pi }{M_{d}}\cdot p_{kM_{d}}}\}$$
where
(36)$$p_{kx}\left\{ \begin{array}{l} \ell_{k}+x-1,\;\;\;\;x\in [1,M_{d})\\ \ell_{k}+\delta_{e}-1,\;\;\;x=M_{d} \end{array}\right.$$
and
(37)$$\ell_{k}=\frac{M_{d}}{2\pi }\arg s_{0}(k)-\left\lfloor \frac{M_{d}}{2\pi }\arccos(1-\frac{\xi^{2}}{2})\right\rfloor$$
and
(38)$$\delta_{e}\left\{ \begin{array}{l} 1+2\left\lfloor \frac{M_{d}}{2\pi }\arccos(1-\frac{\xi^{2}}{2})\right\rfloor ,\;\;\xi \in [0,2)\\ M_{d},\;\;\;\;\;\;\;\;\;\;\;\;\;\;\;\;\;\;\;\;\;\;\;\;\;\;\;\;\;\;\;\;\xi =2\;\; \end{array}\right.$$
Therefore, (34) can be rewritten as:(39)$$\varphi_{k}\in \frac{2\pi }{M_{d}}[p_{k1},p_{k2},\ldots ,p_{kx},\ldots ,p_{kM_{d}}]$$
The continuous phase transmit waveform design problem (33) is transformed into a discrete phase transmit waveform design problem, and the expression is as follows:(40)$$\left\{ \begin{array}{l} \underset{\varphi_{k}}{\max}\;\;\;\frac{r_{n1k}+r_{n2k}\cdot e^{j\cdot \varphi_{k}}}{r_{d1k}+r_{d2k}\cdot e^{j\cdot \varphi_{k}}}\\ \;\;s.t.\;\;\;\varphi_{k}\in \frac{2\pi }{M_{d}}[p_{k1},p_{k2},\ldots ,p_{kx},\ldots ,p_{kM_{d}}],\\ \;\;\;\;\;\;\;\;k=1,2,\ldots ,N_{t}L \end{array}\right.$$

It can be seen that problem (33) and (40) are a one-dimensional function of the phase of the waveform element. The optimal solution can be found through the existing search algorithm. We use the optimization toolbox in MATLAB to quickly solve $\varphi_{k}$. Synthesize the waveform $s{\left(k\right)}^{\left(n\right)}$ calculated this time by (16) and replace $s{\left(k\right)}^{\left(n-1\right)}$ of the previous iteration. Therefore, the constant envelope transmit waveform obtained after the kth calculation in the nth iteration is expressed as:(41)$$s^{\left(k,n\right)}={\left[s{\left(1\right)}^{\left(n\right)},s{\left(2\right)}^{\left(n\right)},\ldots ,s{\left(k\right)}^{\left(n\right)},s{\left(k+1\right)}^{\left(n-1\right)},\ldots ,s{\left(N_{t}L\right)}^{\left(n-1\right)}\right]}^{T}$$
where $s{\left(k+1\right)}^{\left(n-1\right)}$ represents the waveform element that has not yet been calculated, and the value carried is the waveform value of the (n−1)th iteration. 

We discuss the monotonicity of the algorithm JD-CD, assuming:(42)$$SINR(\varphi_{k})=\frac{r_{n1k}+r_{n2k}\cdot e^{j\cdot \varphi_{k}}}{r_{d1k}+r_{d2k}\cdot e^{j\cdot \varphi_{k}}}$$

According to the above analysis, we know the calculation of problem (33) or (40) and the signal expression (16) to synthesize the transmit waveform $s^{\left(k,n\right)}$, and we obtain:(43)$$\begin{array}{ll} SINR{\left(\varphi_{N_{t}L}\right)}^{\left(n-1\right)} & \leq SINR{\left(\varphi_{1}\right)}^{\left(n\right)}\leq SINR{\left(\varphi_{2}\right)}^{\left(n\right)}\;\;\;\leq \ldots \\ & \leq SINR{\left(\varphi_{k}\right)}^{\left(n\right)}\leq SINR{\left(\varphi_{k+1}\right)}^{\left(n\right)}\leq \ldots \\ & \leq SINR{\left(\varphi_{N_{t}L}\right)}^{\left(n\right)} \end{array}$$
where $SINR{\left(\varphi_{N_{t}L}\right)}^{\left(n-1\right)}=SINR^{\left(n-1\right)}$ represents the SINR calculated from the $s(N_{t}L)$ waveform element during the (n−1)th iteration. The transmit waveform at this time is $s^{\left(N_{t}L,n-1\right)}=s^{\left(n-1\right)}$. Additionally, $s^{\left(N_{t}L,n\right)}=s^{\left(n\right)}$, the expression of $s^{\left(n\right)}$ is:(44)$$s^{\left(n\right)}={\left[s{\left(1\right)}^{\left(n\right)},s{\left(2\right)}^{\left(n\right)},\ldots ,s{\left(k\right)}^{\left(n\right)},s{\left(k+1\right)}^{\left(n\right)},\ldots ,s{\left(N_{t}L\right)}^{\left(n\right)}\right]}^{T}$$

It should be noted that each calculation of SINR is optimized again on the basis of the previous iterative calculation. Therefore, the monotonicity of the system output SINR is guaranteed. In addition, complete the nth iteration design and obtain s(n). In the next iteration, the receive filter w(n+1) is updated through the MVDR technique (Equation (12)), and then the first optimization in the (n+1)th iteration is performed through the optimization problem (33) or (40) and signal expression (16) to design s(1,n+1).
**Algorithm 2.** JD-CD**Input**: Nt,Nr,Md,θ0,α02,{αi2,θi}i=1K,αv2,ξ.**Output**: sopt and wopt0: Set n=0, initialize the transmit waveform s(n)=s0 (use expression (13) to calculate the reference waveform s0, s0=vec(S0)) and the receive filter w(n) is calculated by (12). Then, compute SINR(n) by (10).1: n=n+1,k=02: s(n)=s(n−1)3: Compute w(n) by (12)4: Compute Rnum and Rden by (24) and (25), respectively5: k=k+16: Compute rn1k and rn2k by (28) and (29), respectively7: Compute rd1k and rd2k by (31) and (32), respectively8: Compute φk(n) using the problem in (33) or (40)9: Synthesize s(k)(n) by (16) and to replace s(k)(n−1) immediately10: If k=NtL, s(n)=s(NtL,n) and compute SINR(φNtL)(n) by (42),   then SINR(n)=SINR(φNtL)(n). Otherwise, return to step 5.11: If |SINR(n)−SINR(n−1)|≤η, where η is a user selected parameter to control convergence, stop iterating the output optimal transmit waveform sopt=s(n) and receive filter wopt=w(n). Otherwise, repeat step 1 until convergence.

Finally, we analyze the computational complexity of the JD-CD algorithm. The overall complexity of JD-SFR is linearly related to the number of iterations and the length of the waveform s. In particular, it is required to calculate $r_{n1k}$, $r_{n2k}$, $r_{d1k}$ and $r_{d2k}$ in each iteration. The complexity of this step is $O(2{\left(LN_{t}\right)}^{2})+O(2(LN_{t}))$. The complexities of $r_{n1k}$ and $r_{d1k}$ are both $O({\left(LN_{t}\right)}^{2})$; the complexities of $r_{n2k}$ and $r_{d2k}$ are both $O(LN_{t})$. Therefore, the overall computational complexity is $O(n_{o}{\left(LN_{t}\right)}^{3})+O(n_{o}{\left(LN_{t}\right)}^{2})$.

## 5. Simulation Results

In this section, we evaluate the performance of the proposed algorithms (JD-SFR and JD-CD) through the following simulations. The simulation verifies the influence trend under strong signal-dependent interferences from the aspects of SINR, signal-to-noise ratio (SNR), and interference-to-noise ratio (INR), and observes the interference suppression effect of the proposed algorithms through the beampattern, and finally verifies the degree of satisfaction of the designed waveform in terms of amplitude and phase. The MIMO radar using JD-SFR and JD-CD in this paper is compared with the traditional phased array radar, the method in [23] (Joint Optimization of Transmit waveform and Receive filter by Sequential Algorithm, JOSA) and the method in [47] (Joint Optimization Algorithm Case, Majorization-Maximization Algorithm for Parametric Problem, JOAC-MAPP). It should be noted that we will collectively refer to JOSA, JOAC-MAPP and PA as other methods in the following chapters. Optimization problems (22) can be effectively solved with the convex optimization toolbox CVX [58] in MATLAB. Table 1 shows the specific parameters of the simulations.

In the first simulation, the iterative output SINR of the proposed algorithms in the MIMO radar and other methods is compared, and the interference suppression performance under different interference numbers is verified by the beampattern.

Figure 2 shows the optimized SINR results of other methods and our proposed algorithms in this paper under the same conditions. It can be seen from Figure 2 that the SINR has been significantly improved after the joint optimization design of the transmit waveform and the receive filter. The proposed algorithm 1 (JD-SFR) only takes two iterations to complete the convergence. The SINR output by JD-SFR and proposed algorithm 2 (JD-CD) is higher than that of JOSA and PA and JOAC-MAPP, but neither of them converges faster than JOAC-MAPP.

Figure 3 and Figure 4 show the normalized output beampattern after waveform optimization under different signal-dependent interference numbers. Figure 3 shows the output beampattern when K=3. It can be seen from Figure 3 that the transmission powers of JD-SFR, JD-CD and JOSA have good focusing effects in the target direction, and both can form a null in the interference azimuth, while the main lobe of PA and JOAC-MAPP output beampattern is somewhat offset. We proposed method JD-SFR to form a deeper null at azimuth −50°, −10° and 40° (−74.17 dB, −52.55 dB and −48.09 dB, respectively), and more accurate suppression of multiple interferences. As the number of interferences increases, Figure 4 depicts the beampattern for six interferences. It can be seen from Figure 4 that the beampattern output by JOSA, JD-SFR and JD-CD produces nulls near the direction of the arrival of six interferences. JOAC-MAPP forms the deepest null (−102.5 dB) at $\theta_{3}=-30{}^{\circ}$, but it cannot effectively suppress other interference ($\theta_{i\neq 3},i=1,2,\ldots ,K$) and the output sidelobe level is high. We have observed that there are only three effective nulls formed in PA, and the nulls at the 85° azimuth angle are invalid, which is far from achieving the effect of suppressing six interferences at the same time. The reason is that based on the concept of sum co-array [59,60,61], when the number of antennas is $N_{t}=N_{r}=4$, the phased array radar has $N_{t}=4$ degrees of freedom, and can only suppress three interference signals at most. However, for the full-rank covariance matrix, MIMO radar has $2\times N_{t}-1=7$ degrees of freedom, it can suppress at most $2\times N_{t}-2=6$ interferences. Therefore, in the face of the six interferences in Figure 4, the traditional phased array radar cannot completely suppress these interferences.

In the second simulation, Figure 5 and Figure 6 show the SINR level for all methods in the presence of three interferences and six interferences versus the interference-to-noise ratio. It can be seen from Figure 5 that when the INR increases, the output SINR of the proposed method JD-CD has a small loss (0.17 dB) in the range from −10 dB to 40 dB. We observe that the output SINR loss becomes larger (>2.6 dB) after the INR of JD-SFR, JOSA, JOAC-MAPP and PA exceeds 20 dB. Figure 6 shows that the output SINR by PA and JOAC-MAPP drops significantly (18.94 dB and 10.31 dB, respectively) in the high INR range under six interferences. However, the output SINR of JD-SFR has only a small decrease. 

The minimum azimuth angle difference between the target and the interference is defined as follows:
(45)$$\Delta \theta_{\min}=\min\{|\theta_{0}-\theta_{i}|,\;i=1,2,\ldots ,K\}$$

In the third simulation, Figure 7 and Figure 8 show the numerical relationship between SINR and SNR from −30 dB to 30 dB, and analyze the angle locations between the target and the three interferences as farther ($\Delta \theta_{\min}\geq 25{}^{\circ}$) and closer ($\Delta \theta_{\min}\leq 10{}^{\circ}$). Suppose that a point target is located at 15° or 50°, there are three interferences and parameter settings as shown in Table 1. As can be seen from Figure 7, compared with the other methods, the output SINR of the proposed JD-CD is the highest. As we can see in Figure 8, when $\Delta \theta_{\min}$ is reduced to 10°, all methods have different sizes of decline, but the output SINR by our proposed JD-SFR and JD-CD is still very large compared to other methods.

In the fourth simulation, Figure 9 and Figure 10 show the amplitude and phase curves of the transmit waveform optimized by constant envelope and similarity constraints under different similarity parameters. As can be seen in Figure 9, even if different similarity parameters are set, the values of transmit waveforms optimized by JD-SFR and JD-CD are all distributed on the circle with radius c, indicating that the waveforms meet the expected effect and have constant envelope property. A constant envelope waveform can make the radar transmitter work at saturation state and exert its maximum efficiency. Next, we verify that the waveform designed under similarity constraints is similar to the LFM signal. Figure 10 compares the phase relationship between the designed waveforms in this paper and the reference signal when similarity parameters ξ are 0.1, 0.5, 1.5, and 2, respectively. As can be seen from Figure 10, the smaller the similarity parameters, the smaller the gap between the designed waveform and the LFM signal, indicating that the similarity constraints in this paper can effectively carry out phase constraints on the designed waveform. In other words, the waveform meets the expected effect.

## 6. Conclusions

In this paper, we propose two algorithms to jointly design the transmit waveform and receive filter to reduce the influence of interference on the target detection performance of the MIMO radar system and improve the output SINR. We also consider two important practical constraints for the designed waveforms: the constant envelope constraint and the similarity constraint. JD-SFR solves the established non-convex optimization problem by using SDR and fractional programming transformation and obtains a high precision approximate solution to the original waveform optimization problem by randomization. JD-CD decomposes the optimization problem from the aspect of phase, and gradually synthesizes each waveform element based on gradient descent. The simulation results show that our proposed two algorithms have a higher output SINR than other methods, and can effectively suppress the interference according to the prior information. In the presence of strong power interference, the SINR loss of the proposed algorithms is the smallest compared with other methods, and the designed waveform achieves the expectation of constant envelope and similarity. The disadvantage of this paper is that JD-SFR complexity is higher than other methods in this paper, and the SINR of JD-CD is more susceptible to interference than JD-SFR. The following work needs to be improved. 

We intend to study in the future mainly multi-weak targets and multi-strong interferences in harsh mountain environments, assuming that the prior information is unknown, and designing robust waveforms with good waveform characteristics and low complexity.

## Figures and Tables

**Figure 1 sensors-21-03887-f001:**
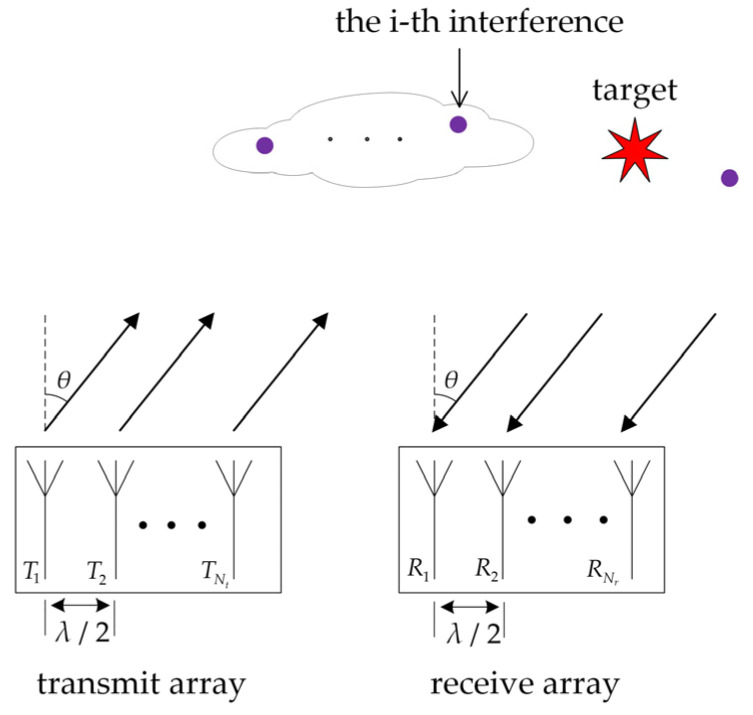
Illustration of the MIMO radar with uniform linear arrays.

**Figure 2 sensors-21-03887-f002:**
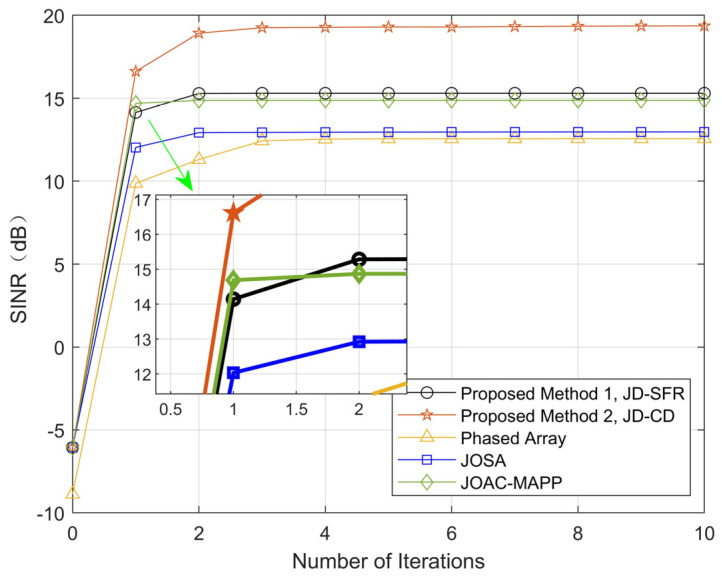
Compare the output SINR of the proposed methods and other methods.

**Figure 3 sensors-21-03887-f003:**
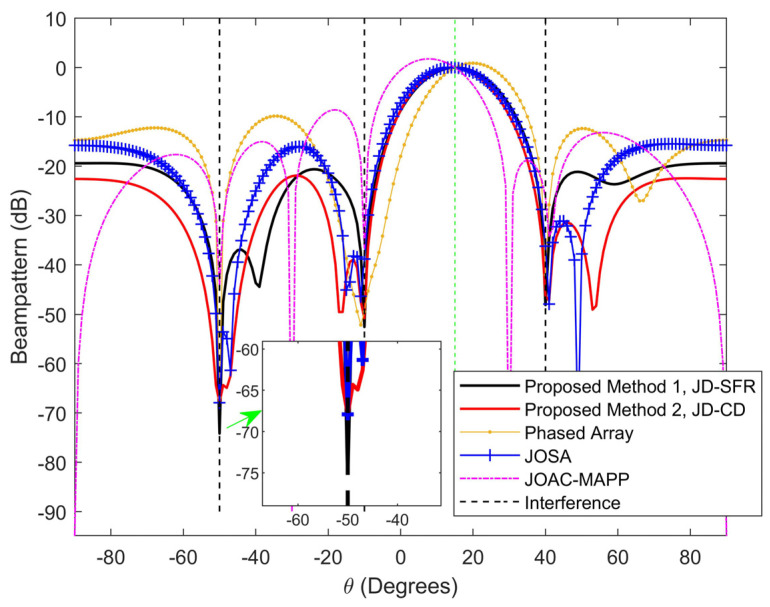
Figure 3. Compare the beampattern of the proposed methods and other methods from three different interference azimuth angles at [−50°,−10°,40°].

**Figure 4 sensors-21-03887-f004:**
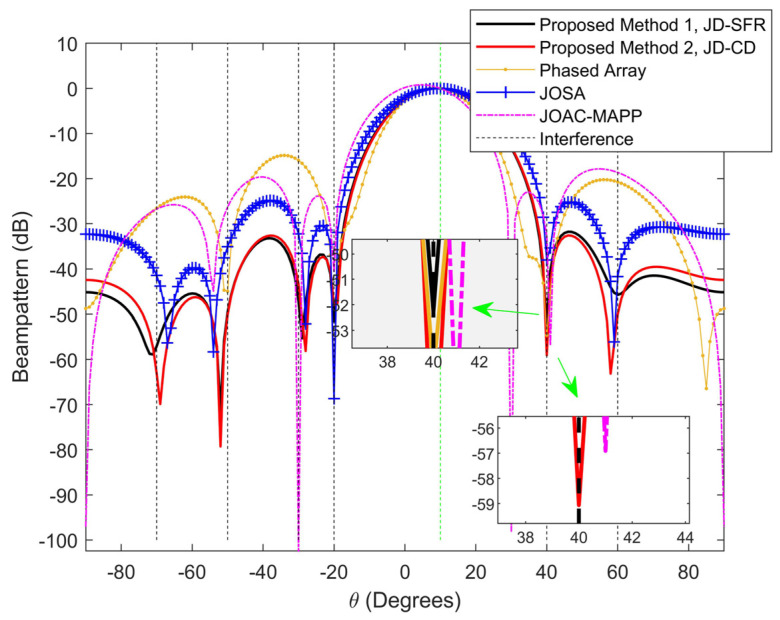
Compare the beampattern of the proposed methods and other methods from six different interference azimuth angles at [−70°,−50°,−30°,−20°,40°,60°].

**Figure 5 sensors-21-03887-f005:**
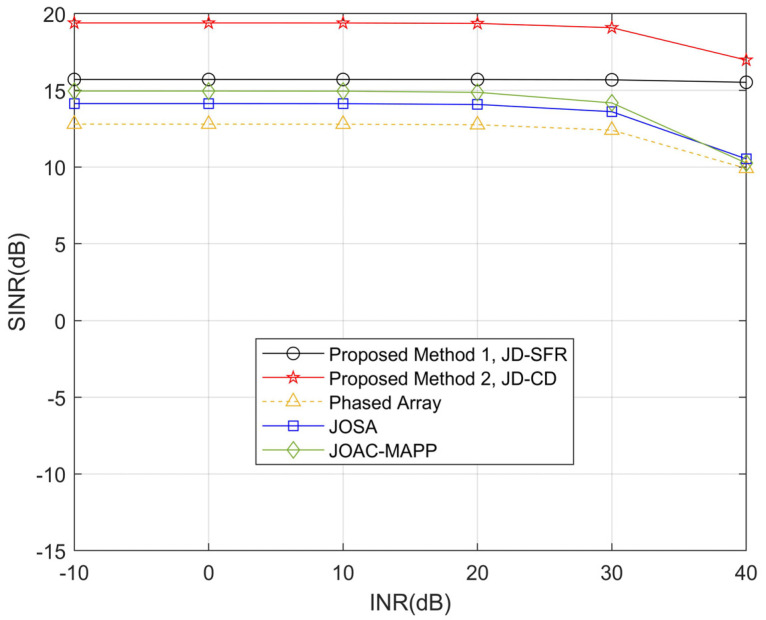
Compare the relationship between SINR loss and interference power obtained by the proposed methods and other methods for *K* = 3.

**Figure 6 sensors-21-03887-f006:**
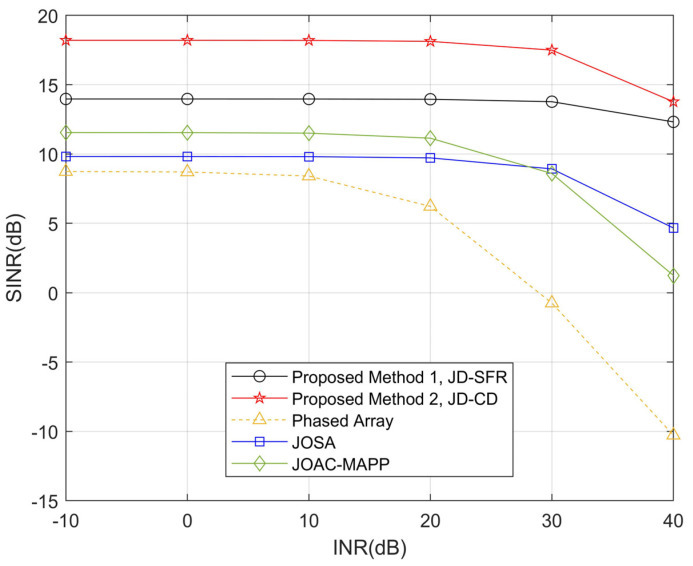
Compare the relationship between SINR loss and interference power obtained by the proposed methods and other methods for *K* = 6.

**Figure 7 sensors-21-03887-f007:**
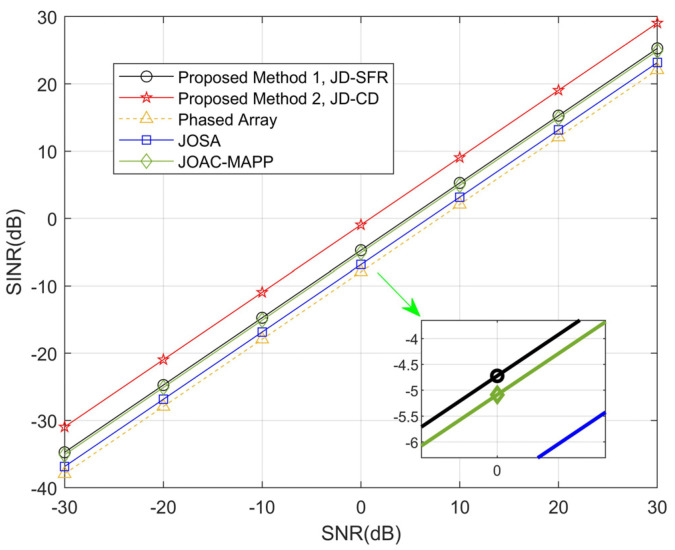
Comparison of obtained SINR using the proposed methods and other methods for Δ*θ*_min_ = 25°.

**Figure 8 sensors-21-03887-f008:**
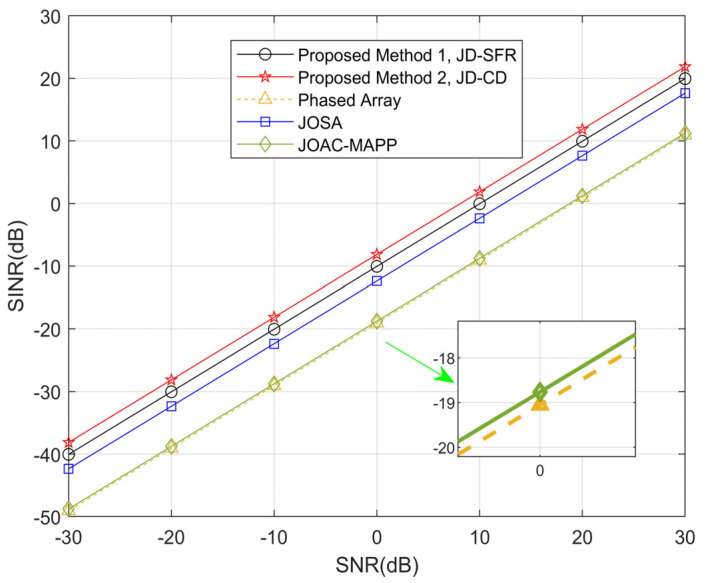
Comparison of obtained SINR using the proposed methods and other methods for Δ*θ*_min_ = 10°.

**Figure 9 sensors-21-03887-f009:**
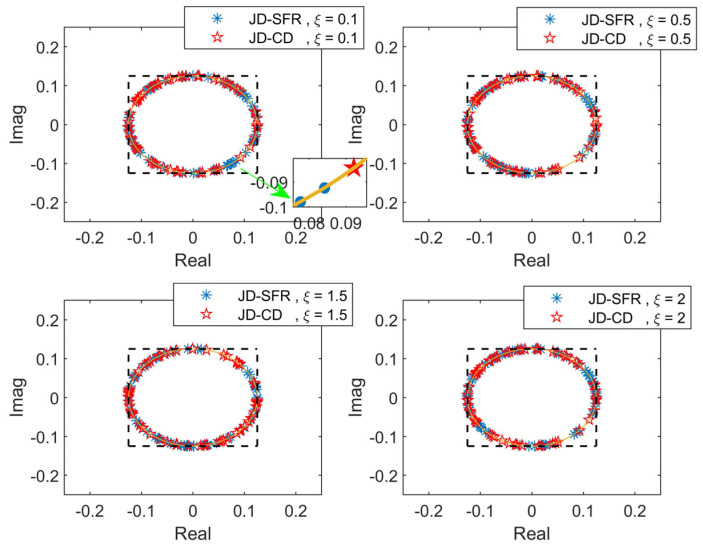
Amplitude characteristics of designed waveforms under different similarity parameters.

**Figure 10 sensors-21-03887-f010:**
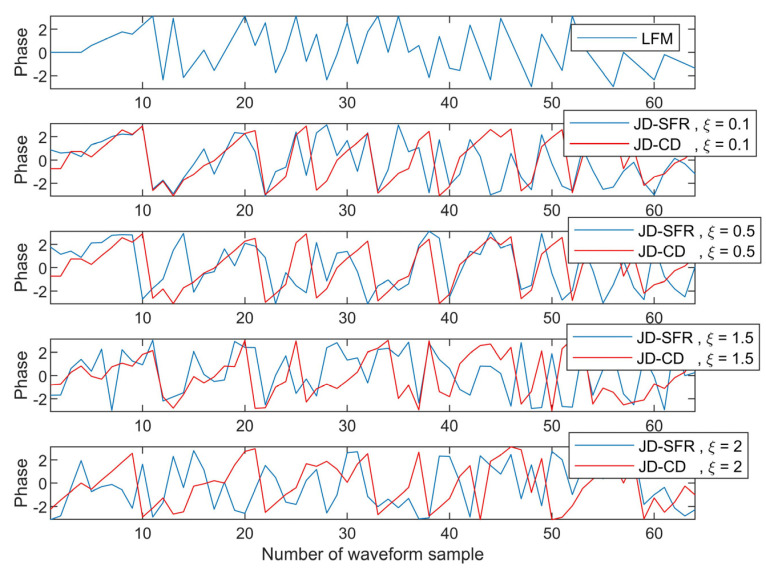
The phase difference between designed waveform and LFM under different similarity coefficients.

**Table 1 sensors-21-03887-t001:** System simulation parameters.

Simulation Parameter	First Simulation	Second Simulation	Third Simulation	Fourth Simulation
Arrangement spacing of ULA: d	λ/2	λ/2	λ/2	λ/2
$N_{t}$	4	4	4	4
$N_{r}$	4	4	4	4
L	16	16	16	16
M	300	300	300	300
$M_{d}$	32	32	32	32
ξ	0.5	0.5	0.5	[0.1, 0.5, 1.5, 2]
K	3	3	3	3
	6	6		
$\theta_{0}$	15°	15°	15°	15°
	10°	10°	50°	
$\theta_{i}$	[−50°,−10°,40°]	[−50°,−10°,40°]	[−50°,−10°,40°]	[−50°,−10°,40°]
	[−70°,−50°,−30°,−20°,40°,60°]	[−70°,−50°,−30°,−20°,40°,60°]		
$\alpha_{0}^{2}$	20 dB	20 dB	[−30, 30] dB	20 dB
$\alpha_{i}^{2},i=1,2,\ldots ,K$	20 dB	[−10, 40] dB	20 dB	20 dB
$\sigma_{v}^{2}$	0 dB	0 dB	0 dB	0 dB
η	0.001	0.001	0.001	0.001

## Data Availability

Not applicable.

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
