# Peer review of "Joint Design of Colocated MIMO Radar Constant Envelope Waveform and Receive Filter to Reduce SINR Loss"

_sensors, 2021, doi:10.3390/s21113887_

Round 1
Reviewer 1 Report
1- Some language edit is required before publication. For example, it is written in Algorithm 1, "by Gaussian randomization technology" where Gaussian randomization is not a "technology", it can be considered as a "technique".
2- Please note that a similar technique based on CD for this problem is already addressed in the following papers:
[a] - M. M. Feraidooni, D. Gharavian, M. Alaee-Kerahroodi and S. Imani, "A Coordinate Descent Framework for Probing Signal Design in Cognitive MIMO Radars," in IEEE Communications Letters, vol. 24, no. 5, pp. 1115-1118, May 2020, doi: 10.1109/LCOMM.2020.2971210.
[b] - M. M. Feraidooni, M. Alaee-Kerahroodi, S. Imani and D. Gharavian, "Designing Set of Binary Sequences and Space-Time Receive Filter for Moving Targets in Colocated MIMO Radar Systems," 2019 20th International Radar Symposium (IRS), 2019, pp. 1-10, doi: 10.23919/IRS.2019.8768146.
[c]- M. Alaee-Kerahroodi, S. Imani, M. R. Bhavani Shankar, M. M. Nayebi and B. Ottersten, "A Coordinate Descent Framework to Joint Design of MPSK Sequences and Receive Filter Weights in MIMO Radar Systems," 2019 IEEE Radar Conference (RadarConf), 2019, pp. 1-6, doi: 10.1109/RADAR.2019.8835518.
Author Response
Dear Reviewers,
Thank you for your comments concerning our manuscript entitled “Joint Design of Colocated MIMO Radar Constant Envelope Waveform and Receive Filter to Reduce SINR Loss” (Manuscript ID: sensors-1240965).
Those comments are all valuable and very helpful for revising and improving our paper, as well as the important guiding significance to our researches. We have studied comments carefully and have made correction which we hope meet with approval. Revised portion are marked in red in the paper. The main corrections in the paper and the responds to the reviewer’s comments are as flowing:
Responds to the reviewer’s comments:
Reviewer #1:
- Response to comment: (Some language edit is required before publication.)
Response: Language changes have been made in the manuscript as suggested.
- Response to comment: (similar technique based on CD in the following papers. [a, b, c])
Response: After studying these three documents [a, b, c], they all used the CD framework in the process of designing the waveform, and they also achieved excellent results ([a,b,c] Already added to our reference list). However, a careful study will reveal that these three documents and my article have their own characteristics.
For example, in [a,b,c], although the authors are all improving the SINR, the constraints imposed by the literature [a] (PSL and ISL) are completely different from our article; in [b,c], the design Discrete phase waveforms only consider the constant envelope characteristics.
Considering comprehensively, the common feature of the methods proposed in [a,b,c] is the design of discrete-phase transmit waveforms and receive filters based on the CD framework. The JD-CD method proposed by us is suitable for both discrete waveforms and continuous phase waveforms. In addition, the waveform designed in this paper is similar to the LFM signal with good waveform characteristics in addition to satisfying the constant envelope. In short, the method proposed in our article has its own differences and applicability characteristics.
Special thanks to you for your good comments. We look forward to your reply.
Reviewer 2 Report
Although the present work provides an interesting contribution to waveform and receive filter design for co-located MIMO radar, it is apparent that the following major comments should be addressed by the authors before publication:
1) Sec. I – I would rephrase the sentence “Recently, a new radar system called multiple-input multiple-output (MIMO) [1] radar has attracted the attention of many researchers” as MIMO radar has now become a quite mature technology.
2) Although the discussion of related literature is satisfactory, it would be very useful if the authors could add a complementary table which categorizes the reviewed works along their main distinctive characteristics so as to better position the present work by difference. Additionally, the following related works on waveform-receive filter design have been missed by the authors:
"Robust waveform and filter bank design of polarimetric radar." IEEE transactions on aerospace and electronic systems 53.1 (2017): 370-384
"Theory and application of optimum transmit-receive radar." Record of the IEEE 2000 International Radar Conference [Cat. No. 00CH37037]. IEEE, 2000.
3) Sec. I – Please rephrase the sentence “In this paper, we consider that in the detection environment of multiple strong signal- 151 dependent interferences.” aiming at improved English.
4) Sec. I – Statement of contributions: “ Compared with other methods, the beampattern by our proposed method has deeper nulls in the interference direction and good energy accumulation in the target azimuth. “ -> Please mention the references/baselines to which the authors are referring from the beampattern standpoint. The same applies to SINR and other features described in the bullet list.
5) When discussing literature on co-located multistatic/MIMO radar, the following closely-related works dealing with the design/analysis of sophisticated receive filters (considering also interaction among the scatterers/targets) have been missed:
"Performance analysis of time-reversal MUSIC." IEEE Transactions on Signal Processing 63.10 (2015): 2650-2662.
"Time-reversal imaging with multiple signal classification considering multiple scattering between the targets." The Journal of the Acoustical Society of America 115.6 (2004): 3042-3047.
6) Due to the large number of acronyms involved, it would be useful providing a table collecting all them.
7) I would refrain the authors from using “Algorithm 1” and “Algorithm 2”, but rather using acronyms which are indicative of the algorithms’ rationales (as done later in the paper).
8) Sec. 2 – The authors consider ULAs in their model assumption. I would like them to clarify whether this is a limiting assumption for the considered design.
9) Please discuss the computational complexity involved in the proposed JD-SFR and JD-CD algorithms.
10) Sec. 5 – Please collect all the parameters concurring to describe the considered simulation setup in a table for reader’s convenience.
11) I recommend that the authors add an illustrative figure to support the discussion of the considered system model.
12) At the end of Sec. 5 (“Conclusion”), please provide an appropriate paragraph which clearly highlights future directions of research.
Author Response
Dear Reviewers:
Thank you for your comments concerning our manuscript entitled “Joint Design of Colocated MIMO Radar Constant Envelope Waveform and Receive Filter to Reduce SINR Loss” (Manuscript ID: sensors-1240965).
Those comments are all valuable and very helpful for revising and improving our paper, as well as the important guiding significance to our researches. We have studied comments carefully and have made correction which we hope meet with approval. Revised portion are marked in red in the paper. The main corrections in the paper and the responds to the reviewer’s comments are as flowing:
Responds to the reviewer’s comments:
Reviewer #2:
Response to comment: (rephrase the sentence “Recently,…researchers”)
Response: The sentence has been paraphrased as “Multiple-input multiple-output (MIMO) [1] radar has attracted the attention of many researchers in the past decades”.
Response to comment: (add a complementary table. Additionally, works on waveform-receive filter design: 370-384; Cat. No. 00CH37037)
Response: In the appendix, a literature analysis table is added. The two missing papers have been added in the proper place of the article.
3. Response to comment: (Sec. I- rephrase the sentence ” In this paper,… ” aiming at improved English.)
Response: The sentence has been paraphrased as “In this paper, we consider a detection environment with multiple strong signal - dependent interferences”.
4. Response to comment: (Sec. I – Statement of contributions: “Compared with other …” mention the references)
Response: Statement of contributions, "other methods" were corrected as "the methods proposed in [23, 47] and PA."
5. Response to comment: (When discussing literature on co-located multistatic/MIMO radar, the following closely-related works have been missed: 2650-2662; 3042-3047)
Response: The related literature on the interaction between scatterers and targets has been re-added to the introduction.( Literature 2650-2662 and 3042-3047 were added to the relevant work [48,49]).
6. Response to comment: (large number of acronyms, add an abbreviation table)
Response: A table of abbreviations was added to the end of the manuscript
7. Response to comment: (I would refrain the authors from using “Algorithm 1” and “Algorithm 2”, but rather using acronyms which are indicative of the algorithms’ rationales)
Response: “Algorithm 1” and “Algorithm 2” have been substituted for JD-SFR and JD-CD in the manuscript.
8Response to comment: (Sec. 2 – The authors consider ULAs … whether this is a limiting assumption)
Response: ULAs are not a limiting assumption. In addition to ULAS, it can also be assumed as Planar Array or Volumetric Array according to the requirements of the scene. Normally, for the convenience of derivation, we assume that it is the signal model of ULAs (which can be considered a special case of Volume Arrays). In general, for the convenience of derivation, we assume that it is a signal model of ULAs (which can be considered a special case of volume arrays). Because system models ultimately derive the SINR function of the design variables (S and W).
9. Response to comment: (discuss the computational complexity)
Response: The computational complexity analysis is located in the algorithm table of the manuscript.
10. Response to comment: (Sec. 5 –Please collect all the parameters concurring to describe the considered simulation setup in a table for reader’s convenience.)
Response: System simulation parameter table (Table 1) is added in Section 5 (5. simulation results).
11. Response to comment: (add an illustrative figure)
Response: The system illustrations figure (Figure 1) is added in Section 2 (2. System model).
12. Response to comment: (provide an appropriate paragraph which clearly highlights future directions of research)
Response: The future research direction paragraph is added after the paragraph of conclusion (6. Conclusion).
We tried our best to improve the manuscript and made some changes in the manuscript. These changes will not influence the content and framework of the paper. And here we did not list the changes but marked in red in revised paper.
Special thanks to you for your good comments. We look forward to your reply.
Reviewer 3 Report
The paper proposed a joint design of co-located MIMO radar constant envelope waveform and receive filter to reduce SINR loss. Numerical results supported the proposed method. Please find the following comments to improve the quality of the paper. 1) In Fig. 9, it is not clear what is the next sample after the right-most part of the waveform. Are they periodic? 2) The presentation of the paper is poor, especially all the figure results are vague. Please check throughout the paper since there were also a lot of typos. The font types of the Equation and the References seem not to match that of the main manuscript.Author Response
Dear Reviewers:
Thank you for your comments concerning our manuscript entitled “Joint Design of Colocated MIMO Radar Constant Envelope Waveform and Receive Filter to Reduce SINR Loss” (Manuscript ID: sensors-1240965).
Those comments are all valuable and very helpful for revising and improving our paper, as well as the important guiding significance to our researches. We have studied comments carefully and have made correction which we hope meet with approval. Revised portion are marked in red in the paper. The main corrections in the paper and the responds to the reviewer’s comments are as flowing:
Responds to the reviewer’s comments:
Reviewer #3:
- Response to comment: (In Fig. 9, it is not clear what is the next sample after the right-most part of the waveform. Are they periodic?)
Response: s is not periodic. The reasons are as follows:
Fig. 9 shows the phase of each element (Nt*L = 64) of the waveform s designed specifically based on the prior information of the scatterers (targets and interferences) in the radar airspace at a certain moment. Specifically, if the information of any of the scatterers changes at the next moment, the waveforms need to be re-designed with the latest prior information (repeat the waveform design process in this paper).
- Response to comment: (The presentation of the paper is poor, especially all the figure results are vague.)
Response: As shown in the newly submitted manuscript. We have substantially revised the presentation in the manuscript and replaced the blurred resulting images with "high fidelity" clear images.
- Response to comment: (Please check throughout the paper since there were also a lot of typos)
Response: We tried our best to improve the manuscript, correct typos, and make some changes to the manuscript. These changes will not influence the content and framework of the paper. We did not list the changes here but marked them in red in the revised paper.
- Response to comment: (The font types of the Equation and the References seem not to match that of the main manuscript)
Response: The formulas and references were all replaced with the same "Palatino Linotype" font as the main manuscript.
Special thanks to you for your good comments. We look forward to your reply.
Round 2
Reviewer 2 Report
The authors have satisfactorily addressed my previous comments and modified their manuscript accordingly. Hence, I am glad to recommend the present work for publication.
This manuscript is a resubmission of an earlier submission. The following is a list of the peer review reports and author responses from that submission.